# Nordic Diet and Inflammation—A Review of Observational and Intervention Studies

**DOI:** 10.3390/nu11061369

**Published:** 2019-06-18

**Authors:** Maria Lankinen, Matti Uusitupa, Ursula Schwab

**Affiliations:** 1Institute of Public Health and Clinical Nutrition, University of Eastern Finland, 70211 Kuopio, Finland; matti.uusitupa@uef.fi (M.U.); ursula.schwab@uef.fi (U.S.); 2Department of Medicine, Endocrinology and Clinical Nutrition, Kuopio University Hospital, 70210 Kuopio, Finland

**Keywords:** Nordic diet, low-grade inflammation, human, diet

## Abstract

Low-grade inflammation (LGI) has been suggested to be involved in the development of chronic diseases. Healthy dietary patterns, such as the Mediterranean diet (MD), may decrease the markers of LGI. Healthy Nordic diet (HND) has many similarities with MD, but its effects on LGI are less well known. Both of these dietary patterns emphasize the abundant use of fruits and vegetables (and berries in HND), whole grain products, fish, and vegetable oil (canola oil in HND and olive oil in MD), but restrict the use of saturated fat and red and processed meat. The aim of this narrative review is to summarize the results of studies, which have investigated the associations or effects of HND on the markers of LGI. Altogether, only two publications of observational studies and eight publications of intervention trials were found through the literature search. Both observational studies reported an inverse association between the adherence to HND and concentration of high sensitivity C-reactive protein (hsCRP). A significant decrease in the concentration of hsCRP was reported in two out of four intervention studies measuring hsCRP. Single intervention studies reported the beneficial effects on interleukin 1Ra and Cathepsin S. Current evidence suggests the beneficial effects on LGI with HND, but more carefully controlled studies are needed to confirm the anti-inflammatory effects of the HND.

## 1. Introduction

Low-grade inflammation (LGI) has been linked with the pathogenesis of several chronic diseases, such as cardiovascular diseases, type 2 diabetes, certain cancers, and neurodegenerative diseases [1,2,3,4]. It is well known that human adipose tissue is not only a storage for excess energy, but it is also an active endocrine organ producing and releasing a number of bioactive compounds, adipokines, of which some are known to be pro-inflammatory (e.g., tumor necrosis factor alfa (TNF-α), Interleukins (IL) 6 and 8), and some anti-inflammatory (e.g., adiponectin) [5,6]. Besides adipose tissue, white blood cells (e.g., monocytes), macrophages, and the liver produce pro- and anti-inflammatory factors [7,8]. It has also been suggested that there is an interaction between the gut microbiota and the immune system, and changes in the gut microbiota may contribute to LGI by resulting in a deficient immune response and impaired tolerance to commensal microorganisms [9,10]. Various mechanisms through which LGI may contribute to the development of non-communicable diseases have been discussed in several reviews [3,6,7,11].

Obesity, especially abdominal obesity and the accumulation of visceral fat, is associated with LGI, and weight loss is the key treatment for LGI in overweight and obese people [12]. Weight loss decreases the elevated concentrations of plasma inflammatory markers and increases e.g., low levels of anti-inflammatory adiponectin [12,13,14,15]. However, the quality of diet and its individual components may also affect the concentrations of inflammatory markers, regardless of weight loss [12]. Especially the Mediterranean diet and some other healthy dietary patterns have been shown to decrease the markers of LGI [16,17]. Those dietary patterns have been characterized as plant food based ones (mostly fruit, vegetables and whole grain, and with a little red meat) [16].

Healthy Nordic diet (HND) has many similarities with the Mediterranean diet [18]. Both dietary patterns emphasize the abundant use of fruits and vegetables, whole grain products, and fish, but will restrict the use of saturated fat (milk fat) and red and processed meat. Olive oil is an important source for unsaturated fat in the Mediterranean diet, whereas canola oil is used in the HND. The HND includes also local berries, like bilberries, lingonberries, and strawberries [18]. There is some variation in the HND between the different Nordic countries and regions, e.g., in consumed fish species and types of berries, fruits, vegetables, and bread.

Recently, Kaluza and colleagues developed a questionnaire-based Anti-Inflammatory Diet Index (AIDI), which could predict systemic chronic inflammation in the Nordic population [19]. It was based on a 123-item food frequency questionnaire among 3503 women with high sensitivity C reactive protein (hsCRP) plasma concentration <20 mg/L. Altogether, 20 foods were statistically significantly related to hsCRP. Foods with anti-inflammatory potential were fruits and vegetables, tea, coffee, whole-grain bread, breakfast cereal, low-fat cheese, chocolate, dried fruits, herbal tea, olive and canola oils, legumes, nuts, linseeds, red wine, and beer. Foods with pro-inflammatory potential were unprocessed red meat, processed meat, organ meat, chips, and soft-drink beverages. This study did not investigate the complete dietary pattern, but hypothesized that these foods might be related to LGI, especially in the Nordic population.

The aim of this narrative review is to summarize the results of studies, which have investigated the associations or the effects of the HND on markers of LGI in observational and intervention studies, respectively. There are several different biomarkers that are used for assessing LGI in dietary studies. In these studies, the markers have been CRP, hsCRP, IL-6, IL-1 Ra, leptin, high-molecular weight adiponectin, TNF-α, IL-1β, IL-10, TNF RII, and Chatepsin S. Furthermore, inflammation on the level of expression of genes that are linked to inflammation in adipose tissue and peripheral blood mononuclear cells (PBMC) is an interesting approach to examine the impact of diet on LGI in the target tissues [20,21].

## 2. Methods

We carried out a literature search of the Nordic diet and LGI and related biomarkers in PubMed. We used the following search term: (Nordic diet OR Baltic sea diet) AND (inflammation OR biomarker OR metabolomics) AND human NOT animal. The search was performed until the 30th of April 2019 and it gave, altogether, 38 publications. Relevant articles were selected based on the review of the abstracts (Figure 1). All of the studies that were related to the topic were included in this review. There were two publications reporting results from observational studies [22,23] (Table 1 and Table 2A) and three publications from intervention studies from two different interventions, SYSDIET [20,24] and New Nordic Diet [25] (Figure 1, Table 1 and Table 2B). In addition, we included three intervention studies that reported the effects of the HND on LGI related markers, but were not encompassed by the search. They were the NORDIET intervention [26,27] and the New Nordic Diet intervention [28]. We also included the SYSDIMET intervention trial, which included three key Nordic food items, i.e., fatty fish, bilberries, and whole grain rye products in line with HND pattern [29] and new publication from the SYSDIET study samples investigating gene-expression in PBMCs [30]. Altogether, there were two publications from three observational studies and eight publications from four intervention studies (Figure 1, Table 1 and Table 2).

## 3. HND and LGI in Observational Studies

Thus far, there are two observational studies (including data from three cohorts) investigating the association between the HND (named as the Baltic Sea Diet in these studies, which corresponds the same as HND) and LGI. Those studies have consistently shown an inverse association between the HND and hsCRP (Table 2A).

The association between the HND and inflammatory markers was independently investigated in two large cross-sectional studies in Finnish participants: in the DIetary, Lifestyle, and Genetic determinants of Obesity and Metabolic syndrome (DILGOM) study (*n* = 4579), and in the Helsinki Birth Cohort Study (HBCS, *n* = 1911) [23]. In both studies, the participants filled in a standardized and validated food frequency questionnaire (FFQ) that was designed to measure the habitual diet over the previous 12 months. Baltic Sea Dietary Score (BSDS), which reflects the HND (score from 0 to 25), was calculated based on the FFQ, and it included nine dietary features: (1) Nordic fruits (apples, pears, and berries); (2) Nordic vegetables (tomatoes, cucumber, leafy vegetables, roots, cabbages, peas); (3) Nordic cereals (rye, oat, and barley); (4) low-fat and fat-free milk; (5) Nordic fish (salmon and freshwater fish); (6) ratio of PUFA to SFA and trans-fatty acids; (7) low intake of red and processed meat; (8) total fat (as % of total energy); and, (9) moderate or low intake of alcohol. The higher the BSDS the better the adherence to the HND. LGI was measured using the following inflammatory markers: leptin, high-molecular weight (HMW) adiponectin, TNF-alfa, IL-6, and hsCRP. In the Determinants of Obesity and Metabolic Syndrome (DILGOM) study, hsCRP concentrations inversely associated with the BSDS (*p* < 0.01) in all adjusted models, which included relevant confounding factors, such as age, sex, energy intake, education, smoking, physical activity, waist circumference, medication for diabetes, and the use of statins. Participants’ mean concentration of hsCRP was 1.14 mg/L in the lowest BSDS quintile and 1.03 mg/L in the highest BSDS quintile. Unexpectedly, HMW-adiponectin concentrations also inversely associated with the BSDS (*p* < 0.05). Adjustments for the above-mentioned confounding factors strengthened this association (*p* < 0.01). Other measured inflammatory markers did not associate with BSDS in the DILGOM study. Alcohol intake was the only component that significantly associated with the HMW-adiponectin concentrations of the single BSDS components. Participants with a high intake of alcohol had higher HMW-adiponectin concentration than individuals with low alcohol intake (*p* < 0.001). This finding is consistent with earlier findings that were related to alcohol consumption and adiponectin [31]. In the HBCS, the hsCRP inversely associated with the BSDS in all adjusted models (*p* < 0.01). Other measured inflammatory markers did not associate with BSDS in the HBCS study. Among the single BSDS components, the higher intake of Nordic fruits, berries, and cereals, and the lower intake of red and processed meat and moderate alcohol intake significantly associated with lower hsCRP concentrations in both of the studies.

Both the DILGOM and the HBCS study were included among the Health 2000 Survey in the meta-analysis investigating associations with BSDS and cardiometabolic risk factors, including hsCRP [22] (Table 2A). In this meta-analysis, the risk of elevated hsCRP concentration was lower both among men (OR 0.58, 95% CI 0.43, 0.78) and women (OR 0.73, 95% CI 0.58, 0.91) in the highest BSDS quintile than among those in the lowest BSDS quintile.

In summary, it has been consistently shown that a high adherence to the HND lowers the risk of low-grade inflammation, but, so far, the studies are scarce and only conducted among Finns.

## 4. Nordic Diet and LGI in Randomized Dietary Trials

### 4.1. Studies Including Selected Key Components of the HND

Only three intervention studies have investigated the effects of the HND pattern on LGI (Table 2B). Furthermore, the effect of the selected key components of the HND on LGI has been studied in several randomized dietary trials. For example, bilberries, which are rich in phenolic compounds [32] and commonly used in the Nordic countries, decreased the inflammatory score, which was calculated based on concentrations of hsCRP, IL-6, IL-12 and LPS, during the 8-week intervention period [33]. We investigated the effect of low insulin response grain products (including rye bread, sourdough whole grain wheat bread, and dark pasta), fatty fish, and bilberries on several inflammatory markers in the SYSDIMET intervention [29]. Altogether, 104 participants with characteristics of metabolic syndrome who had completed the study were analyzed. The study participants were randomized for 12 weeks into one of three dietary groups: (1) a Healthy Diet group (included whole grain and low insulin response grain products, fish and bilberries), (2) Whole Grain Enriched Diet group (WGED), and (3) control diet group (included refined cereal products and limited fish and berry consumption). HsCRP, TNF-α, IL-6, IL1Ra serum amyloid A (SAA), chemokine (C-C motif) ligand 5 (CCL5), soluble intercellular cellular adhesion molecule-1 (sICAM-1), and macrophage migration inhibitory factor (MIF) were measured as markers of LGI before and after the intervention. There were no significant differences in the changes in these inflammatory markers, except in hsCRP in the participants, who did not use statins during the intervention. The plasma hsCRP concentrations decreased in individuals following the WGED and Healthy Diet interventions (*p* < 0.01 and *p* < 0.05, respectively) and the change in hsCRP in the WGED group was significantly different from that in the control group (*p* < 0.05) (Figure 2). Interestingly, after the intervention, hsCRP was at the same level as in statin users, who had low levels already in the onset of the study (Figure 2).

The effect of a prudent low-fat breakfast (based on Nordic foods) as compared with usual breakfast was studied in a parallel controlled 12-week study with 79 healthy overweight [34]. The prudent breakfast included oat bran porridge with low-fat milk or yogurt, bilberry or lingonberry jam, whole grain bread, low-fat spread (high in PUFA), poultry or fatty fish, and fruit or berries. Foods were provided *ad libitum*. CRP and TNF-R2 was significantly decreased by the prudent breakfast when compared with the control breakfast (*p* < 0.005). The percentual changes in CRP and TNF-R2 between baseline and 12 weeks were +37% for the control breakfast and −30% for the prudent breakfast, and +9% for the control breakfast and +2% for the prudent breakfast, respectively. There were no changes in body weight, but sagittal abdominal diameter, which is a marker of visceral fat, was reduced in the prudent breakfast group.

### 4.2. HND in Controlled Dietary Trials

The effect of the HND pattern on LGI has been investigated in three randomized controlled trials (Table 2B). In the Dietary Intervention With Shop Model (SHOPUS), the objective was to investigate whether the *ad libitum* New Nordic Diet (NND) versus Average Danish Diet (ADD) in adults (18–65 y) with central obesity and components of the metabolic syndrome could reduce body weight and improve the risk markers of the metabolic syndrome, type 2 diabetes, and cardiovascular diseases [25,28]. The NND included foods as fruits and vegetables (especially berries, cabbages, root vegetables, and legumes), potatoes, fresh herbs, wild plants, and mushrooms, nuts, whole grain, meats from livestock and game, fish and shellfish, and seaweed [35]. The control ADD was the diet that was typically eaten by the adult Danish population. The decrease in energy intake and weight loss was greater in the NND group as compared with the ADD group. The plasma CRP concentration decreased in the NND group (*p* = 0.007) [27]. The decrease of CRP attenuated, but remained significant after adjusting for weight loss (*p* = 0.043). TNF-α was measured for the subset of the study population (*n* = 64), and it did not change significantly [25]. It is worth knowing that these beneficial results were not sustained in the follow-up of the study [36]. However, higher compliance with NND after the active intervention was associated with less body weight regain [36].

In the NORDIET trial, 88 normal to slightly overweight hyperlipidemic men and women were randomly assigned to *ad libitum* Nordic prudent diet or a control (habitual) diet for six weeks [26,27]. Altogether, 86 participants completed the study with measurements of circulating CRP and cathepsin S levels. Similarly as in the SHOPUS study, the body weight decreased in the HND group as compared with the controls, despite the *ad libitum* nature of the HND diet. Cathepsin S is a proteolytic enzyme that has been associated with LGI [37,38]. Compared with a habitual control diet, the HND decreased cathepsin S levels (*p* = 0.03). However, the decrease was not significant after adjusting for change in body weight and LDL-cholesterol concentration, which may indicate that the decrease of the level of S cathepsin was mediated by weight loss or lowered LDL cholesterol concentration. There were no changes in CRP (*p* = 0.40) [26].

In the SYSDIET study [24], altogether 200 middle-aged individuals with characteristics of metabolic syndrome and impaired fasting glucose/glucose intolerance were randomized into a HND group or a control group for 18 to 24 weeks in six study centers in Finland (2), Sweden (2), Denmark (1), and Iceland (1). The diets were isocaloric in order to keep body weight unchanged. HND included whole-grain products, local berries, fruits and vegetables, canola oil, three fish meals per week and low-fat dairy products. Compliance to diet was monitored by repeated food diaries and serum phospholipid fatty acid profile [24], and later on by alcylresorcinols (fiber intake) and serum beta-carotene (intake of vegetables) [39]. There were more drop-outs in the control group (27% vs. 7.9%), and 96 subjects in the HND group and 70 subjects in the control group completed the study. Some differences in responses to dietary changes were found across different study centers, but, there was an improved lipid profile with HND when compared to control group, and lipid changes and blood pressure reduction were more beneficial in most compliant participants of the HND group when examined in more detail in relation to compliance. No significant changes were found in the weight or glucose tolerance between the study groups. Among the various inflammatory markers examined (hsCRP, IL-1beta, IL-1Ra, IL-6, IL-10, and TNF RII, HMW adiponectin), only IL-1Ra showed a different response between the diets: it increased in the control group while it remained stable in the HND, and at the end of the study a marked difference was seen in IL-1Ra between the groups. The high intake of saturated fatty acids associated with an increase in IL-1Ra, whereas magnesium intake was inversely related to IL-1Ra, but the association with dietary fiber intake was not significant [24]. These results on IL-1Ra may be of importance, since IL-1Ra is a sensitive marker of insulin resistance and it has been shown to predict type 2 diabetes [40]. HND tended to increase plasmalogens based on the further analyses of the serum samples from the SYSDIET study, while a decreasing trend was found in ceramides in line with anti-oxidative and anti-inflammatory properties of HND [41]. Furthermore, in two sub-studies of the SYSDIET, down-regulations of immune response related genes were reported in adipose tissue [20] and PBMCs [30]. Recently, we found associations between dietary fiber intake, serum indole propionate coming from gut, inflammation, and insulin secretion capacity in the study participants of the Finnish Diabetes Prevention Study followed over 10 years. This suggests an intimate role of dietary fiber in the regulation of glucose metabolism and LGI [42,43].

## 5. Discussion

A Mediterranean type diet, which is much more studied when compared to Nordic diet, has consistently shown to be anti-inflammatory [17]. Mediterranean and Nordic diets have a lot of common features, including abundant use of fruits and vegetables (and berries in the Nordic Diet), the restricted use of saturated fats (milk fat) and red meat, the use of vegetable oil (olive oil in the Mediterranean diet and canola oil in the Nordic diet), and whole grain products as an important source of dietary fiber [19]. They both emphasize the consumption of fish. Consumed fish species are a bit different, but the fat quality and quantity is about the same. In the Nordic countries, fish species, such as salmon, rainbow trout, Baltic herring, mackerel, saithe, and cod, are consumed. Olive and canola oils differ a bit according to their fatty acid content. Canola oil has lower amounts of saturated and monounsaturated fatty acids and more n-3 and n-6 polyunsaturated fatty acids when compared with olive oil. Despite some differences in the diets, this review suggests that the HND may also have anti-inflammatory effects. Adherence to the Mediterranean diet has not been easy in other parts of the Western world [44]. Therefore, it is important to consider local food culture in health promoting food patterns to help in adherence to the diet. Similar to variations in Mediterranean diet by country, there is also some variation in the HND between the different Nordic countries and regions, e.g., in consumed amount and type of fish, berries, fruits, vegetables, and bread. For example, Finns consume more berries and whole grain rye as compared with others [45], whereas Icelanders and Norwegians consume more fish. Therefore, the HND is not exactly the same among the reported studies and in multicenter study, such as SYSDIET, there is also variation in the diet within the study.

Overall, currently, it is not possible to draw definite conclusions of anti-inflammatory effects of HND, since there are quite few studies investigating this issue. Furthermore, several different markers of inflammation have been applied in the studies. It might be seen as a strength that favorable effects have been seen in the circulating markers of inflammation as well as in gene expression level in adipose tissue and PBMCs [20,30]. Recently, Sakhaei et al. published a systematic review and meta-analysis of randomized controlled clinical trials with regard to the effects of the HND pattern on circulating inflammatory markers. [46]. They concluded that adherence to the Nordic diet pattern does not seem to affect circulating CRP, IL-6, and TNF-α. However, this kind of meta-analysis may have limitations due to different study designs and adherence to diets. Other inflammatory markers were not studied in the meta-analysis. The meta-analysis also included a trial in which HND was used as a control diet for a Paleolithic-type diet, which caused greater weight loss when compared with HND [47,48]. Regarding the study design, that study [47,48] did not really measure the effect of HND on LGI.

All of the studies investigating the health effects of HND, and reported in this review, were conducted in the Nordic countries, which enables a reasonable compliance to the HND in the intervention groups. The control diet was usually the average diet in the Nordic countries. However, as seen e.g., in the SYSDIET study [24], the people volunteering to these studies are generally more aware of the principles of the healthy diet as average people are. Therefore, the change in diet is not very dramatic in the intervention groups and the control participants may need to worsen their diet. Overall, the change to the HND from a more westernized diet means that the nutrient concentration of the diet becomes higher and the diet becomes closer to the current Nordic recommendation for both the nutrient and food intake level [49]. This happened indeed e.g., in the SYSDIET study [24]. Better adherence to HND still increased the differences between HND and the average control diet in the SYSDIET study [40].

One limitation in these studies is that there was a significant weight loss during the intervention in some of these [26,28]. These studies do not make it possible to examine the effects of the dietary composition as such without concomitant weight changes. However, change in body weight may be considered as a natural part of the outcome, as changing to a healthier diet may simultaneously decrease the energy intake and stimulate weight loss. Weight loss makes it harder to interpret the results, but it could be considered in statistical analyses. In some studies, positive effects on inflammatory markers have been seen without weight loss [24,29] or a decrease in CRP has remained significant after adjusting for the weight loss [28]. Another possible weakness in these studies is that LGI has not been a main primary outcome of many of these studies, but still were considered as primary outcomes. Thus, the studies might have been underpowered for LGI, and many participants had low levels of LGI already at the baseline. There are some variations in the characteristics of the participants among the studies e.g., in age, Body Mass Index (BMI), and health status (Table 1), which also may explain the variation in the results. Furthermore, the baseline diet and compliance to the HND might have been varied between the studies.

## 6. Concluding Remarks

This narrative review overviews and summarizes all the published papers covering the HND and LGI. Only 10 papers describing seven studies were found, including three observational studies and four interventions. The authors are not aware of other reviews investigating this issue in detail. Quite few studies have investigated the effect of HND on LGI so far. Two observational studies have shown an inverse association between the adherence to the HND and concentration of hsCRP. In the intervention studies, a significant decrease in the concentration of hsCRP has been reported in two studies [28,29] out of four studies measuring hsCRP [24,26,28,29]. Furthermore, the HND has had beneficial effects on concentrations of other inflammatory markers, such as IL-1Ra [24] and Cathepsin S [27], and it has been shown to downregulate the gene expression of inflammation related genes in adipose tissue and PBMC [20,30]. It is noteworthy that these positive effects have been seen without significant weight loss. These results suggest that the HND may have anti-inflammatory effects, but more carefully controlled studies are needed to confirm the anti-inflammatory effects of the HND.

## Figures and Tables

**Figure 1 nutrients-11-01369-f001:**
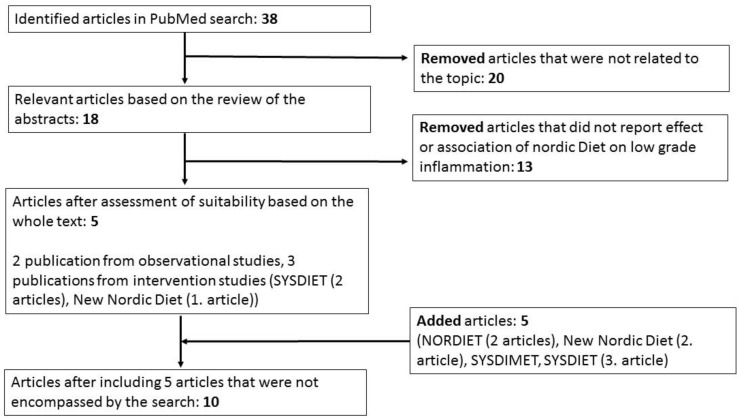
Procedure for selection of articles.

**Figure 2 nutrients-11-01369-f002:**
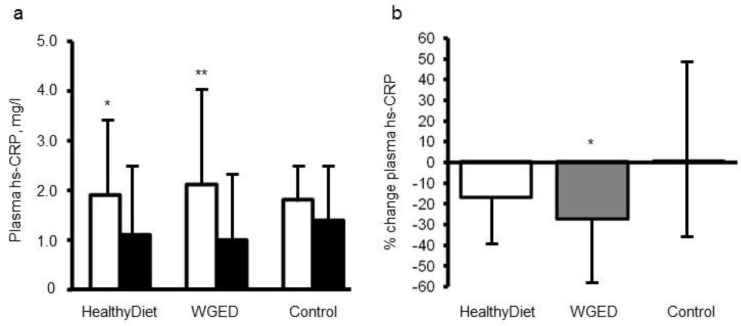
Plasma concentrations (**a**) and relative changes (**b**) of high sensitivity C-reactive protein (hsCRP) according to study group in the participants not using statins during the SYSDIMET intervention [29]. (**a**) White bars represent hsCRP concentrations at baseline and black bars after a 12 week consumption of HND (*n* = 27), Whole Grain Enriched Diet (WGED) (*n* = 24) or control diet (*n* = 25).* *p* < 0.05 and ** *p* < 0.001 for baseline vs. week 12 in Student’s paired test. (**b**) *p* = 0.04 for the group effect in general linear model univariate analysis. * *p* < 0.05 for the difference between WGED and control groups after Bonferroni correction for multiple comparisons. Values are median and interquartile range (IQR).

**Table 1 nutrients-11-01369-t001:** Characteristic of studies investigating association or effect of the Healthy Nordic diet (HND) on low-grade inflammation (LGI).

References	Years of Data Collection	Study Design	Name of the Study	Country	Population
[22,23]	2007	Observational study	DILGOM	Finland	*n* = 4579 A representative sample of the Finnish population in 5 large study areas
[22,23]	2001–2004	Observational study	the Helsinki Birth Cohort Study	Finland	*n* = 1911 Helsinki University Central Hospital area
[22]	2000–2001	Observational study	Health 2000 Survey	Finland	*n* = 5180 A representative sample of the Finnish population from 80 health service districts
[20,24,30]	2009–2010	RCT multicenter 18–24 weeks	SYSDIET	Denmark Finland Iceland Sweden	*n* = 166 men and women with features of MetS mean age 55 years mean BMI 31.6 kg/m^2^
[25,28]	2010–2011	RCT 26 weeks	New Nordic Diet	Denmark	*n* = 147 centrally obese men and women mean age 42 years mean BMI 30.2 kg/m^2^
[26,27]	2007–2008	RCT 6 weeks	NORDIET	Sweden	*n* = 86 mildly hypercholesterolaemic men and women mean age 53 years mean BMI 26.5 kg/m^2^
[29]	2008–2009	RCT 12 weeks	SYSDIMET	Finland	*n* = 104 men and women with features of MetS mean age 59 years mean BMI 31.1 kg/m^2^

DILGOM, The Dietary, Lifestyle and Genetic Determinants of Obesity and Metabolic Syndrome study; MetS, Metabolic syndrome; RCT, Randomized controlled trial, BIM, Body Mass Index.

**Table 2 nutrients-11-01369-t002:** Observational studies investigating association between the Nordic diet and inflammatory markers (**A**) and intervention studies investigating effects of the Nordic Diet on inflammatory markers (**B**).

	**Reference**	**Study Name**	**Dietary Intake**	**Inflammatory Markers/Measurements Related to Inflammation**	**Main Results**
**(A)**	[23]	DILGOM study (*n* = 4 579); Helsinki Birth Cohort Study (*n* = 1911)	FFQ was used to measure dietary intake over the past year and to calculate the BSDS.	leptin, HMW-adiponectin, TNF-alfa, IL-6, and hsCRP.	An inverse association between the BSDS and hsCRP concentration in both studies (*p* < 0.01). No association with other inflammatory markers. In the DILGOM study HMW-adiponectin had inverse association with BSDS.
	[22]	DILGOM (*n* = 4776); Health 2000 Survey (*n* = 5180); Helsinki Birth Cohort Study (*n* = 1972)	FFQ was used to measure dietary intake over the past year and to calculate the BSDS.	hsCRP	The risk of elevated hsCRP concentration was lower among men (OR 0.58, *p* = 0.004) and women (OR 0.73, *p* = 0.001) in the highest BSDS quintile than among those in the lowest BSDS quintile.
	**Reference**	**Study Name**	**Study Groups (*n*) and Duration of the Intervention**	**Inflammatory Markers/Measurements Related to Inflammation**	**Main Results**
**(B)**	[24]	SYSDIET 6 study centers in Nordic countries	(1) HND (*n* = 96) (2) Control (average Nordic diet) (*n* = 70) 18–24 weeks	IL-1Ra, IL-1β, IL10, TNF RII, hsCRP	IL-1 Ra increased in the Control group. No differences between the groups in the other markers.
	[20]	SYSDIET (3 centers out of 6)	(1) HND (*n* = 31) (2) Control (*n* = 25) 18–24 weeks	Gene expression in subcutaneous adipose tissue	Gene expression of inflammation related genes was reduced in the HND group compared with the Control group.
	[30]	SYSDIET (3 centers out of 6)	(1) HND (*n* = 42) (2) Control (*n* = 26) 18–24 weeks	Gene expression in peripheral blood mononuclear cells	Pathways and processes involved in the immune response were down-regulated in the HND group.
	[28]	New Nordic Diet	(1) NND (*n* = 91) (2) ADD (*n* = 56) 26 weeks	CRP	CRP decreased in the NND group (*p* = 0.007). The decrease of CRP attenuated, but remained significant after adjusting for the weight loss (*p* = 0.043). The loss of body weight during the intervention was greater (*p* < 0.001) in the NND group (~4.74 ± 0.48 kg) than in the ADD group (~1.52 ± 0.45 kg).
	[25]	Subset of New Nordic Diet	(1) NND (*n* = 43) (2) ADD (*n* = 21) 26 weeks	CRP, TNF-α	No significant changes in the whole population, but in women CRP concentration decreased 40% in the NDD group (*p* < 0.01). The model was not adjusted by weight loss. The loss of body weight during the intervention was greater in NDD than ADD (*p* < 0.01).
	[26,27]	NORDIET	(1) HND (*n* = 44) (2) Control (*n* = 42) 6 weeks	CRP, Cathepsin S	No change in CRP. Level of Cathepsin S was decreased in the HND group compared with the Control group (*p* = 0.003). The difference remained significant after adjusting for baseline Cathepsin S level, but not after adjusting for change in weight or LDL cholesterol concentration.
	[29]	SYSDIMET	(1) Healthy Diet rich in whole grain, fatty fish and bilberries (2) Whole Grain Enriched Diet (WGED) (3) Control Diet 12 weeks	hsCRP, TNF-α, IL-6, IL1Ra, SAA, CCL5, sICAM-1 and MIF	Plasma hsCRP concentration decreased in the WGED and Healthy Diet groups (*p* < 0.01 and *p* < 0.05, respectively) and the change in hsCRP in the WGED group was significantly different from that in the control group (*p* < 0.05). No changes in other inflammatory markers.

ADD, average Danish diet; BSDS, Baltic See Dietary Score; CCL5, chemokine (C-C motif) ligand 5; CRP, C-reactive protein; FFQ, food frequency questionnaire; HND, Healthy Nordic Diet; HMW-adiponectin, high molecular weight adiponectin; hsCRP, high sensitivity C-reactive protein; IL, interleukin; IL1Ra, Interleukin 1 receptor antagonist; MIF, macrophage migration inhibitory factor; NND, new Nordic Diet, SAA, serum amyloid A; sICAM-1, soluble intercellular cellular adhesion molecule-1; TNF-α, tumor necrosis factor alfa; TNF RII, tumor necrosis factor receptor II, WGED, Whole Grain Enriched Diet.

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
