# Peer review of "Nordic Diet and Inflammation—A Review of Observational and Intervention Studies"

_nutrients, 2019, doi:10.3390/nu11061369_

Reviewer 1 Report

Review for nutrients-493507

Overall: This paper was a systematic review researching the impact of a Nordic diet on low-grade inflammation. For the most part, studies showed associations between the Nordic diet and lower inflammation levels. Although I don’t have anything against a review of the Nordic diet and inflammation, there were limitations that limit the current importance of such a paper. First, I am concerned with the search terms, see my specific comments below. The other major limitation is that there is no synthesis of the results. Synthesis of results goes beyond simply reporting on the results of the study. How do the studies compare to each other? What were the limitations of the previous studies that may limit their interpretability? What are the major future recommendations for research or clinical practice that make following a Nordic diet important? These are the types of questions that need to be addressed in a Discussion section of a systematic review.

Abstract:

1.     Low-grade should be hyphenated throughout.

2.     Can you specify somewhere in the abstract that this is a systematic review?

3.     I would change “literature survey” to “literature search”.

4.     The phrase “both epidemiological studies reported….” is confusing to me. I am assuming you are only referring to the two observational studies. Intervention trials are still epidemiological work.

5.     Need to spell out hsCRP.

Introduction:

6.     Change “inflammatory” to “pro-inflammatory” in the second sentence of the introduction.

7.     The sentence starting with “Besides adipose tissue……...” needs to be rewritten. You don’t need the word “also” if you are using the word “besides” already. Also, “blood white cells” should be “white blood cells”.

8.     I understand the text about weight loss and obesity in the beginning of the second paragraph, but are they really needed? The objective of the study isn’t to look at LGI and obesity. In fact, you state that quality of diet can influence inflammatory markers regardless of weight loss or obesity. I think you could just start the second paragraph with that. The text on weight loss almost drives a reader to think this review is focused on weight-loss interventions.

9.     I also am not sure of the point of the paragraph on the AIDI. Another tool, the Dietary Inflammatory Index has been used in Nordic populations and has a much larger literature-base related to prediction of LGI. I am not saying you need to mention the Dietary Inflammatory Index, but why single out the AIDI when other dietary inflammatory-based indices exist? Also, that paragraph seems out of place, not sure what it has to do with seeing if a dietary pattern is associated with LGI. Based on what I can tell, the AIDI is a lot like the Dietary Inflammatory Index in that it is not a dietary pattern in and of itself, it is a way to characterize dietary patterns. However, you are focused on a specific dietary pattern.

10.  Are there any other differences between the Nordic and Mediterranean diet since you are highlighting this comparison? I would imagine the fish being consumed is quite different. I am guessing the Nordic diet has more cold-water fish, correct?

11.  Can you specifically note in the last paragraph of the introduction that this was a systematic review?  

Methods

12.  I would change “literature survey” to “literature search”.

13.  Change the word “hits” to “publications”.

14.  What searching database did you use? I don’t see that mentioned. Also, I strongly urge trying the search and removing the “AND human”. If you used something like PubMed, I sometimes find that not all studies are tagged for human or animal. This may cause you to miss studies.

15.  Please clarify the sentence starting with “There were two publications…….”. Again, intervention studies are still epidemiological studies, so I am not sure what you mean by this.

16.  My point about the search terms may be indicated by the three studies included not found by your search. How did you come across them if they weren’t in the search? That needs to be mentioned if you don’t redo your search.

17.  Does SYSDIET, SYSDIMET, or NORDIET stand for something?

18.  In table 1, the footnotes need to indicate what all acronyms for the studies mean.

Results:

19.  For the BSDS, are higher scores indicative of more adherence?

20.  What did the control diet group get in the SYSDIMET study?

21.  I don’t see how the paper by Ulven and colleagues is relevant? It looks like it examined several specific dietary components that may be common to a Nordic diet, but they are common to lots of diets? Was this included simply because it used a Nordic population? I disagree with this being a study on a Nordic dietary pattern and should probably be removed from the study. I can see the argument for the other two studies in section 4.1, but this one is a stretch.

Author Response

Reviewer 1

Overall: This paper was a systematic review researching the impact of a Nordic diet on low-grade inflammation. For the most part, studies showed associations between the Nordic diet and lower inflammation levels. Although I don’t have anything against a review of the Nordic diet and inflammation, there were limitations that limit the current importance of such a paper. First, I am concerned with the search terms, see my specific comments below. The other major limitation is that there is no synthesis of the results. Synthesis of results goes beyond simply reporting on the results of the study. How do the studies compare to each other? What were the limitations of the previous studies that may limit their interpretability? What are the major future recommendations for research or clinical practice that make following a Nordic diet important? These are the types of questions that need to be addressed in a Discussion section of a systematic review.

Ø  We thank the reviewer for this important comment. We have added to the discussion section some future recommendations for clinical practice and emphasize some other practical issues.

We would like to highlight that this manuscript is a review, and it does not fulfil all of the criteria for a systematic review. We did not find enough studies on this topic to be able to perform a systematic review. We have included all the studies we found in our search related to the topic, without grading the publications or prior selection.

Abstract:

1.     Low-grade should be hyphenated throughout.

Ø  Low-grade have been hyphenated throughout.

2.     Can you specify somewhere in the abstract that this is a systematic review?

Ø  It is important to note that this is not a systematic review although the literature search was comprehensive.

3.     I would change “literature survey” to “literature search”.

Ø  “Literature survey” was changed to “literature search”.

4.     The phrase “both epidemiological studies reported….” is confusing to me. I am assuming you are only referring to the two observational studies. Intervention trials are still epidemiological work.

Ø  To be more clear, we changed “epidemiological studies” to “observational studies”.

5.     Need to spell out hsCRP.

Ø  hsCRP has been spelled out.

Introduction:

6.     Change “inflammatory” to “pro-inflammatory” in the second sentence of the introduction.

Ø  That has been changed as suggested

7.     The sentence starting with “Besides adipose tissue……...” needs to be rewritten. You don’t need the word “also” if you are using the word “besides” already. Also, “blood white cells” should be “white blood cells”.

Ø  The sentence has been rewritten.

8.     I understand the text about weight loss and obesity in the beginning of the second paragraph, but are they really needed? The objective of the study isn’t to look at LGI and obesity. In fact, you state that quality of diet can influence inflammatory markers regardless of weight loss or obesity. I think you could just start the second paragraph with that. The text on weight loss almost drives a reader to think this review is focused on weight-loss interventions.

Ø  We think that it is impossible to write about low-grade inflammation and diet without mentioning obesity and weight loss, which are key factors affecting low-grade inflammation. Therefore, we would like to keep those two sentences regarding this issue, but we tried to modify text so that it should not drive a reader to think that this review is focused on weight loss.

9.     I also am not sure of the point of the paragraph on the AIDI. Another tool, the Dietary Inflammatory Index has been used in Nordic populations and has a much larger literature-base related to prediction of LGI. I am not saying you need to mention the Dietary Inflammatory Index, but why single out the AIDI when other dietary inflammatory-based indices exist? Also, that paragraph seems out of place, not sure what it has to do with seeing if a dietary pattern is associated with LGI. Based on what I can tell, the AIDI is a lot like the Dietary Inflammatory Index in that it is not a dietary pattern in and of itself, it is a way to characterize dietary patterns. However, you are focused on a specific dietary pattern.

Ø  We agree with the Reviewer that AIDI is not a dietary pattern. We changed the location of this paragraph a bit and specified that this study did not investigate dietary pattern.

10.  Are there any other differences between the Nordic and Mediterranean diet since you are highlighting this comparison? I would imagine the fish being consumed is quite different. I am guessing the Nordic diet has more cold-water fish, correct?

Ø  We thank the reviewer for this important comment. We have added discussion related to possible differences between the Nordic and Mediterranean diets.

11.  Can you specifically note in the last paragraph of the introduction that this was a systematic review?

Ø  It is important to note that this not a systematic review. We have specified that this is a comprehensive review based on the current publications on the field.

 Methods

12.  I would change “literature survey” to “literature search”.

Ø  This has been changed as suggested.

13.  Change the word “hits” to “publications”.

Ø  This has been changed as suggested.

14.  What searching database did you use? I don’t see that mentioned. Also, I strongly urge trying the search and removing the “AND human”. If you used something like PubMed, I sometimes find that not all studies are tagged for human or animal. This may cause you to miss studies.

Ø  We used PubMed as mentioned in the manuscript.

We thank the suggestion to try the search without “AND HUMAN” term. It gave 52 articles. However, none of the extra articles found were reporting the effect or association of the healthy Nordic diet on markers of LGI.

15.  Please clarify the sentence starting with “There were two publications…….”. Again, intervention studies are still epidemiological studies, so I am not sure what you mean by this.

Ø  We changed “epidemiological” to “observational”.

16.  My point about the search terms may be indicated by the three studies included not found by your search. How did you come across them if they weren’t in the search? That needs to be mentioned if you don’t redo your search.

Ø  Those three studies were not found without “AND human” term either. We know quite well the research done in this area and were able to add those publications, which were not found by the search string.

17.  Does SYSDIET, SYSDIMET, or NORDIET stand for something?

Ø  The acronyms of SYSDIET, SYSDIMET or NORDIET have not been spelled out in the original articles either.

18.  In table 1, the footnotes need to indicate what all acronyms for the studies mean.

Ø  The acronyms of SYSDIET, SYSDIMET or NORDIET have not been spelled out in the original articles either.

Results:

19.  For the BSDS, are higher scores indicative of more adherence?

Ø  Yes it is. That has been clarified in the text.

20.  What did the control diet group get in the SYSDIMET study?

Ø  In the control group, participants were asked to avoid wholegrain products. They were instructed to replace the breads they usually consumed with refined wheat breads, and other cereal products, e.g. porridge or pasta, with low-fiber products. The intake of bilberries was not allowed and consumption of fish was allowed once a week only. This information was included briefly in the revised manuscript.

21.  I don’t see how the paper by Ulven and colleagues is relevant? It looks like it examined several specific dietary components that may be common to a Nordic diet, but they are common to lots of diets? Was this included simply because it used a Nordic population? I disagree with this being a study on a Nordic dietary pattern and should probably be removed from the study. I can see the argument for the other two studies in section 4.1, but this one is a stretch.

Ø  We removed the paper by Ulven et al.

Reviewer 2 Report

NORDIC DIET AND INFLAMMATION

Thank you for the opportunity to read and Review the manuscript titled “Nordic Diet and inflammation”. In this manuscript, the authors reported on beneficial effects of Nordic Diet on anti-inflammatory effects. Although, this manuscript contains numerous stylistic and scientific errors that must be atended to prior to moving forward.

These are my comments:

1.       Title: The authors must identify the manuscript with a epidemiological design. Is a systematic Review or a meta-analyses?

2.       Abstract: The authors does not provide a structured summary including data sources, study elegibility criteria, study appraisal and synthesis methods. Also, the authors should include the implications of key findings.

3.       Abstract (Line 19): The authors write out hsCRP without defining it previously. Yet in line 48  they refer to “high sensivity protein C reactive” as hsCRP. Other examples found: TNF-alfa, and TNF-α. This is rather distracting, and reflects poor editing.

4.       Introduction: The Introduction is written in fragmentary way, the different concepts mentioned should be better described and linked between them. The authors should restructured the Introduction.

5.       Objectives: The authors do not provide an explicit description of questions being adressed with reference to participants, intervention, comparisons, outcomes and study design (Please describe the PICOS strategy).

6.       Methods: The quality of presentation of Methodology should be improved. First, the authors do not mention the design of the study. Second, if it is a systematic Review, they do not indicate if a review protocol exists, where can be accessed, and if it is available. The authors should provide registration information, including registration number (For example: PROSPERO database).

7.       Methods (continued): Third, the elgibility criteria are poor described. The authors should give a rationale description about the elegibility criteria of the studies included. Fourth, The authors mention that “the search was performed 10 th of January 2019” (lines 70-71), However they do not mention the date last searched.

8.       Methods (Continued): Fifth, The authors do not described the summary measures and the methods used for assessing risk of bias of individual studies. A description of these measures should be given.

9.       Results: The authors have not included a Results section. The findings should be better introduced in the manuscript in the specific section mentioned previously. In table 1 the authors presents the characteristics of the studies included. Consider also the inclusion in the table of the following information: years of data collection, number of participants, parameters assessed and methods of assessment.

10.   Discussion: The authors have not included a Discussion section. The authors do not discuss about the limitations of the study and the outcomes level and at review-level.

11.   According to the line 227 “5.Mediterranean diet and LGI vs. Nordic diet and LGI”, the authors concluded that Nordic diet may have antiinflammatory effects” that are similar to Mediterranean diet due to the food included in both dietary patterns are similar. In my opinión withouth a comprehensive analysis, it is not possible to make this affirmation (it was not the aim of the study and the authors did not include no study that compare both dietary patterns).

Author Response

Thank you for the opportunity to read and Review the manuscript titled “Nordic Diet and inflammation”. In this manuscript, the authors reported on beneficial effects of Nordic Diet on anti-inflammatory effects. Although, this manuscript contains numerous stylistic and scientific errors that must be atended to prior to moving forward.

These are my comments:

1.       Title: The authors must identify the manuscript with a epidemiological design. Is a systematic Review or a meta-analyses?

Ø  We would like to highlight that this manuscript is a review, and it does not fulfill all of the criteria for a systematic review. We did not find enough studies on this topic to be able to perform a systematic review. Here we have included all the studies we found in our search related to the topic.

We changed the title “Nordic diet and inflammation – a review of observational and intervention studies”

2.       Abstract: The authors does not provide a structured summary including data sources, study elegibility criteria, study appraisal and synthesis methods. Also, the authors should include the implications of key findings.

Ø  These are not included since this is not a systematic review.

3.       Abstract (Line 19): The authors write out hsCRP without defining it previously. Yet in line 48  they refer to “high sensivity protein C reactive” as hsCRP. Other examples found: TNF-alfa, and TNF-α. This is rather distracting, and reflects poor editing.

Ø  We apologize these errors. These have been corrected.

4.       Introduction: The Introduction is written in fragmentary way, the different concepts mentioned should be better described and linked between them. The authors should restructured the Introduction.

Ø  The introduction has been slightly restructured. We hope that is more fluent now.

5.       Objectives: The authors do not provide an explicit description of questions being adressed with reference to participants, intervention, comparisons, outcomes and study design (Please describe the PICOS strategy).

Ø  Since this article is not a systematic review, these issues are not addressed in the aims. However, these issues are addressed when describing the studies.

6.       Methods: The quality of presentation of Methodology should be improved. First, the authors do not mention the design of the study. Second, if it is a systematic Review, they do not indicate if a review protocol exists, where can be accessed, and if it is available. The authors should provide registration information, including registration number (For example: PROSPERO database).

Ø  Please, see comments above.

7.       Methods (continued): Third, the elgibility criteria are poor described. The authors should give a rationale description about the elegibility criteria of the studies included. Fourth, The authors mention that “the search was performed 10 th of January 2019” (lines 70-71), However they do not mention the date last searched.

Ø  We did not used any eligibility criteria. We reported all the results related to the topic. We have now added a discussion section where we have discussed limitations of these studies.

We updated the dates for the search and rephrased the sentence.

8.       Methods (Continued): Fifth, The authors do not described the summary measures and the methods used for assessing risk of bias of individual studies. A description of these measures should be given.

Ø  This is not a systematic review or meta-analysis.

9.       Results: The authors have not included a Results section. The findings should be better introduced in the manuscript in the specific section mentioned previously. In table 1 the authors presents the characteristics of the studies included. Consider also the inclusion in the table of the following information: years of data collection, number of participants, parameters assessed and methods of assessment.

Ø  We added the years of data collection. Number of participants and parameters assessed are also included in the table 1.

1.     Discussion: The authors have not included a Discussion section. The authors do not discuss about the limitations of the study and the outcomes level and at review-level.

Ø  We thank the reviewer for pointing out this important missing section. Discussion section has been added.

11.   According to the line 227 “5.Mediterranean diet and LGI vs. Nordic diet and LGI”, the authors concluded that Nordic diet may have antiinflammatory effects” that are similar to Mediterranean diet due to the food included in both dietary patterns are similar. In my opinión withouth a comprehensive analysis, it is not possible to make this affirmation (it was not the aim of the study and the authors did not include no study that compare both dietary patterns).

Ø  The conclusion has been changed.

Reviewer 3 Report

The paper is an overview of papers covering The Nordic diet and low grade inflammation - only 8 papers were found which is an important finding in itself and should be mentioned in the conclusion. On the whole, the paper is well written even though at some instances sentences are very long. Also on the first page, it says blood white cells instead of white blood cells. Very few other mistakes and well interpreted paper. 

Author Response

The paper is an overview of papers covering The Nordic diet and low grade inflammation - only 8 papers were found which is an important finding in itself and should be mentioned in the conclusion. On the whole, the paper is well written even though at some instances sentences are very long. Also on the first page, it says blood white cells instead of white blood cells. Very few other mistakes and well interpreted paper. 

Ø  We would like to thank the reviewer of these positive comments.

Round  2

Reviewer 1 Report

1.     In the new text about the AIDI, the word “hypothesis” should be “hypothesize” on line 73.

2.     In regards to my previous comment about the difference between fish species consumed, you mention the fat quality and quantity being about the same, but there are so many more differences about fish intake that could differ and drive differences in inflammatory potential of diet. I think a little more is needed to describe differences in the Nordic and Mediterranean diets. What about oils? I am sure oils used in those populations differ in content as well.

3.     For the new sentence you added on line 95, I don’t see PMBC spelled out anywhere.

4.     In “Both emphasize also consumption of fish”, take out the word “also”.

5.     In “in other parts of the Western word” on page 9 of 14, “word” should be “world”.

6.     I think the language of the of the new paragraph added to the Discussion about the strengths of the HRD are too strong. Especially, the first couple of sentences that use the word or phrase “strong conclusions” and “comparison of results is demanding”.

7.     I also don’t think use of “on the other hand” is correct because that indicates an opposing view. That sentence seems to actually be in line with the previous sentence, so I am not sure what the “other hand” is referring to.

8.     I think there needs to be a much more thorough explanation of potential limitations of the Sakhaei paper. Most meta-analyses use techniques and sensitivity analyses that take into account the limitations of previous studies, such as difference in study designs. If that paper took those things into account and it didn’t change the overall interpretations of those findings, then that isn’t a limitation of the meta-analysis. If anything, it would be a strength that they considered those limitations and addressed them. What were the actual limitations that the meta-analysis cited related to the studies included in the meta-analysis? You mention some of this in your new paragraph, but relatively vaguely.

9.     The paragraph after the one describing the meta-analysis, what results are those referring to? Your findings or the meta-analysis findings.

Author Response

1.     In the new text about the AIDI, the word “hypothesis” should be “hypothesize” on line 73.

- Thank you for noticing this typo. It has been corrected as suggested.

2.     In regards to my previous comment about the difference between fish species consumed, you mention the fat quality and quantity being about the same, but there are so many more differences about fish intake that could differ and drive differences in inflammatory potential of diet. I think a little more is needed to describe differences in the Nordic and Mediterranean diets. What about oils? I am sure oils used in those populations differ in content as well.

-  We added a bit more discussion related to differences in the HND and Mediterranean diets e.g. by listing fish species consumed in Nordic countries and differences in vegetable oils used.  Actually, there are no comparative studies between MD and HND concerning inflammatory markers.

3.     For the new sentence you added on line 95, I don’t see PMBC spelled out anywhere.

- PBMC has been spelled out in the previous chapter.

4.     In “Both emphasize also consumption of fish”, take out the word “also”.

- Word “also” has been removed.

5.     In “in other parts of the Western word” on page 9 of 14, “word” should be “world”.

- Thank you for noticing the typo. It has been corrected.

6.     I think the language of the of the new paragraph added to the Discussion about the strengths of the HRD are too strong. Especially, the first couple of sentences that use the word or phrase “strong conclusions” and “comparison of results is demanding”.

- The expressions in the paragraph have been softened.

7.     I also don’t think use of “on the other hand” is correct because that indicates an opposing view. That sentence seems to actually be in line with the previous sentence, so I am not sure what the “other hand” is referring to.

- “On the other hand” referred the opposing view of the benefits of several different inflammatory markers. However, “on the other hand” has been removed.

8.     I think there needs to be a much more thorough explanation of potential limitations of the Sakhaei paper. Most meta-analyses use techniques and sensitivity analyses that take into account the limitations of previous studies, such as difference in study designs. If that paper took those things into account and it didn’t change the overall interpretations of those findings, then that isn’t a limitation of the meta-analysis. If anything, it would be a strength that they considered those limitations and addressed them. What were the actual limitations that the meta-analysis cited related to the studies included in the meta-analysis? You mention some of this in your new paragraph, but relatively vaguely.

- It is well known that meta-analyses work when they include intervention studies that have a lot of similarities with regard to the duration, study individuals, intervention diets and methods. We strongly believe that this is not the case with trials investigating healthy effects of HND including inflammation as the main outcome measure.  Sakhaei et al. did sensitivity analyses according to age and gender.

The aim of our invited review article was to consider all the published data available. Please, note that gene expression studies - rarely used in any dietary interventions - also suggest beneficial effects. In fact, compliance to experimental diets is rarely analyzed and reported. Therefore, meta-analyses does not work in dietary intervention studies or observational studies when individual studies are heterogeneous.  

9.     The paragraph after the one describing the meta-analysis, what results are those referring to? Your findings or the meta-analysis findings.

- In practice both (our findings and meta-analysis findings), but to be clear we have added “studies reported in this review”.

Reviewer 2 Report

I am grateful that the authors have tried to improve their manuscript. However, I believe that the quality of the manuscript is still poor. 

First, Although the authors mention in the second review round that this study is not a sistematic review, in a narrative review is also necessary to provide a structured summary including data sources, study elegibility criteria, study appraisal and synthesis methods. Also, the authors should include the implications of key findings. 

Second, Synthesis of results goes beyond simply reporting on the results of the study. How do the studies compare to each other? 

Author Response

I am grateful that the authors have tried to improve their manuscript. However, I believe that the quality of the manuscript is still poor. 

-        It is unfortunate that the reviewer see this manuscript poor. However, the literature search is comprehensive and this is the most relevant method to make a review when the literature is not larger. We would like to emphasize that gene expression studies are almost unique with regard to health effects of healthy dietary patterns. In our mind, these results tend to support our conclusions.

First, Although the authors mention in the second review round that this study is not a sistematic review, in a narrative review is also necessary to provide a structured summary including data sources, study elegibility criteria, study appraisal and synthesis methods. Also, the authors should include the implications of key findings. 

We have done a comprehensive literature search on the topic. As mentioned in the manuscript, PubMed was used as a data source. Since we know the current literature very well, we added some studies, which were not encompassed by the literature search. All steps have shown in the Figure 1.

We have presented in the Table 1 all the relevant data from studies on the main topic. We have tried to make a realistic synthesis from the available data. We are well informed of the strengths and weaknesses that are associated with meta-analyses on dietary studies. We decided to go through all the evidence in a very open way. To be honest, sometimes this may be better than just to try to make conclusions from originally mixed data. In the revised version, we have tried to improve our manuscript as the reviewer is suggesting.

Second, Synthesis of results goes beyond simply reporting on the results of the study. How do the studies compare to each other? 

-        As mentioned in the discussion, the comparison of the results is a bit demanding due to different study designs and inflammatory markers used. However, we have tried to add more comparisons of the studies in the text.